# Delirium in Nursing Home Residents: A Narrative Review

**DOI:** 10.3390/healthcare10081544

**Published:** 2022-08-15

**Authors:** Klara Komici, Germano Guerra, Franco Addona, Carlo Fantini

**Affiliations:** 1Department of Medicine and Health Sciences, University of Molise, 86100 Campobasso, Italy; 2Department of Mental Health, ASREM, Antonio Cardarelli Hospital, 86100 Campobasso, Italy

**Keywords:** delirium, nursing home residents, older people

## Abstract

Delirium is an important component of the geriatric syndromes and has been recognized to negatively influence the prognosis of older people in hospital and in a post-acute setting. About 2–5% of older people world-wide live in nursing homes and are characterized by functional impairment, cognitive decline, dementia, comorbidities, and polypharmacotherapy, all factors which influence the development of delirium. However, in this setting, delirium remains often understudied. Therefore, in this narrative review, we aimed to describe the latest evidence regarding delirium screening tools, epidemiology characteristics, outcomes, risk factors, and preventions strategies in nursing homes.

## 1. Introduction

The Latin word *delirare* means to deviate from a straight line, and the term delirium was initially mentioned by Celsus during the first century to describe mental disorders during head trauma or fever [1]. Delirium is a complex neurocognitive disorder characterized by acute modification of attention, awareness, cognitive functions, and behavioral abnormalities caused by an underlying medical condition [2,3]. Delirium results from a combination of several specific factors such as neuroinflammation, oxidative stress, endocrine dysfunction, and circadian rhythm dysregulation, which lead to a breakdown in cerebral network connectivity and failure of interaction between processing sensory signals and motor effectors [4]. The prevalence of delirium varies considerably by setting: the overall prevalence was 23% in a medical setting [5], 35% in palliative care [6], 1–2% in the community, and up to 70% in a long-term care setting [7]. Delirium has been recognized to negatively influence the prognosis of people with acute illness, and a meta-analysis study which investigated the association of delirium with mortality, institutionalization, and dementia, in hospitalized or post-acute care people aged 65 years or older, revealed that delirium is an independent predictor of poor outcome [8]. Other important negative consequences of delirium are related to the emotional distress and burden among healthcare providers and caregivers. Indeed, a study described that the subjective burden that nurses experience when caring for people with delirium was high and the hyperactive/hyperalert subtype was the most challenging to deal with [9]. The abovementioned studies generally did not focus on nursing home residents (NHRs), and little is known about delirium in a nursing home setting.

About 2–5% of older people worldwide live in nursing home settings [10], and NHRs are characterized by functional impairment, cognitive decline, dementia, multi-comorbidities, and polypharmacotherapy [11], factors which influence the development of delirium, increasing the risk of negative outcomes and the emotional distress of older people and healthcare providers. Nevertheless, delirium remains often undetected and underdiagnosed. Many reasons such as lack of education and training in healthcare providers, lack of implementation of validated detection tools, and negative attitudes hinder the detection and implementation of preventive and therapeutic measures. Hence, it is crucial to have in-depth knowledge regarding delirium in NHRs. In this narrative review, we performed a Medline/PubMed search regarding articles on delirium in NHRs, and we discuss the epidemiological characteristics, risk factors, outcomes, detection tools, and prevention strategies.

## 2. Materials and Methods

Two authors (K.K.) and (C.F.) performed a Medline/PubMed search regarding articles on delirium in NHR from inception until 1 April 2022. Combinations of the following free text terms and major medical subject headings were used: “delirium”, “nursing home”, “long-term care”, and “long-term care facilities”. Additional articles were identified by screening the reference lists of studies included in our review. Two authors (K.K.) and (C.F.) collected data regarding study design, characteristics of participants, delirium diagnostic criteria, and main results from the original research articles that were within the scope of this review. Original research articles which satisfied the following criteria were included: NHR study setting and studies reporting data on delirium prevalence or incidence, as well as studies reporting data on delirium risk factors or delirium prevention. The search strategy was limited to articles published in the English language, and data from non-nursing home populations (assisted living facilities, care homes, or senior housing) were excluded. Multiple studies reporting data from the same cohort were considered, if important information regarding other aims of our review was included.

## 3. Review Results

A total of 1745 articles were identified initially, and, after duplicate removal, 1292 articles were identified for title and abstract screening. Review of titles, abstracts, and full articles yielded a final selection of 38 original studies.

### 3.1. Delirium Prevalence and Incidence in Nursing Homes

From our review, we identified 28 [11,12,13,14,15,16,17,18,19,20,21,22,23,24,25,26,27,28,29,30,31,32,33,34,35,36,37,38] articles with relevant data regarding the prevalence of delirium, and eight articles [11,26,39,40,41,42,43,44] described the incidence of delirium. Total population mean age ranged from 76.2 years to 88.5 years, and 46.8% to 90.8% of the participants in the studies were female.

#### 3.1.1. Delirium Prevalence

The prevalence of delirium ranged widely from 1.4% to 70.3%. Diagnostic Statistical Manual of Mental Disorders, third–fifth edition (DSM-III to DSM-V), Confusion Assessment Methods (CAM), Nursing Home Confusion Assessment Methods (NH-CAM), and the Neelon and Champagne Confusion Scale (NEECHAM) were the most frequently used diagnostic criteria and screening tools.

Morichi et al. [30] and Bo et al. [29] described a point prevalence of about 37% in a multicenter Italian observation study where the four As test (4AT) tool was performed for the detection of delirium. In contrast, a cross-sectional study including six nursing homes in Belgium revealed that the prevalence of delirium assessed by the delirium observational screening scale (DOSS) was no more than 15% [31]. A validation pilot study which compared the results of DSM-V criteria with Informant Assessment of Geriatric Delirium (I-AGeD) reported a prevalence of 5.9% and demonstrated that I-AGeD was suitable for the detection of delirium in Swiss NHR [34]. NHRs affected by COVID-19 disease had a high prevalence of delirium from 49.2% to 57.9%, mainly presented as hypoactive delirium [33,38].

#### 3.1.2. Delirium Incidence

The incidence of delirium also ranged widely from 10% to 60%, and the main diagnostic criteria and screening tools applied were DSM-IV, DSM-V, and CAM. Two studies reported higher incidence of delirium among NHR, at 40% and 60% [40,43], compared to others included in our review. The authors suggested that high prevalence of dementia and need for continued care were possible explanations. Table 1 reports the characteristics of the abovementioned studies regarding the prevalence and the incidence of delirium in NHR.

### 3.2. Risk Factors for the Onset of Delirium in NHR

#### 3.2.1. Demographic and Social Risk Factors

Most of the studies were in agreement about risk factors; however, some contradictions were identified. Voyer et al. [19], Zucchelli et al. [35], and Sepulveda et al. [44] reported that age was a significant risk factor for delirium onset in NHR. Other studies included in our review did not describe a significant association regarding age and delirium onset (Table 1). Female gender was identified as a risk factor in one study [39]; in contrast, another study [27] reported that older people with delirium were less likely to be females. Widowhood was also reported as a risk factor.

#### 3.2.2. Comorbidities

Most of the studies concluded that dementia was an independent risk factor with the exception of one study [44], where data analysis after correction for other risk factors failed to find that dementia was associated with delirium, probably because many people living with dementia could have been recently treated for delirium symptoms without a specific diagnosis of delirium. Parkinson’s disease and depression were other comorbidities reported as risk factors. Of interest, diabetes resulted as a protective risk factor, probably because the frequent glycemic control results in closer contact between nurses and older people, which may avoid the development of delirium [41].

#### 3.2.3. Malnutrition

Nutritional status and other elements that may reflect malnutrition such as low lean muscle mass, dehydration, and electrolyte imbalance have been described as significant predisposing and/or precipitating risk factors [13,15,17,25,35]. Other indicators of malnutrition such as low albumin or low protein level were higher among older people who developed delirium compared to them without but without reaching statistical significance [17].

#### 3.2.4. Infections

Infections were characterized by higher odds of delirium onset [28]. Furthermore, urinary and respiratory infections were mentioned as important precipitating risk factors [25,39]. Delirium was also described in NHRs with SARS-CoV-19 infection [33]. Furthermore, a case study series study reported that, the day after COVID-19 vaccination, delirium or subsyndromal delirium of mild to moderate severity was identified in 10% of NHRs with no potential competing explanation [42]. 

#### 3.2.5. Drugs

Regarding drugs, anticholinergics, antipsychotics, antidepressants and benzodiazepines were most frequently associated to delirium [15,28,30,40,41,43].

#### 3.2.6. Other Factors

Hearing deficits, visual impairment, and functional dependency resulted as significant and independent risk factors in several studies. Chair and physical restraints increased the odds for delirium development [26,44]. Falls showed a strong association with the onset of delirium in NHR [13,41]. Data from a retrospective cohort study including 1571 persons from 12 nursing homes found pain to be significantly associated with delirium [40].

Table 1 summarizes the characteristics of studies that described significant risk factors associated with delirium.

### 3.3. Adverse Outcomes Associated with Delirium in NHRs

An overall increased risk for mortality in NHRs which experienced delirium was reported by different studies [15,21,23,27,36]. In contrast, one study [25] did not find an association between delirium and overall mortality, but other factors such as infections, comorbidities, advanced age, and low plasma albumin level resulted as independent predictors. Rehospitalizations were higher among older people with delirium [27], and an important functional decline was reported as a consequence of delirium [21,27]. Of note, delirium was associated with long-term cognitive decline and dementia development [39]. Studies reporting information regarding the impact of delirium regarding outcomes in NHR are summarized in Table 1.

### 3.4. Delirium Screening Tools

The diagnosis of delirium should be guided by the standard criteria of DSM-V and International Classification of Diseases, 10th Revision (ICD-10). DSM-V criteria for delirium include acute and fluctuating disturbance of attention and awareness, disturbance in cognition, the absence of a pre-existing neurocognitive disorder, and the presence of medical conditions, withdrawal, exposure to toxins, or multiple etiologies [45]. ICD-10 criteria for delirium, similar to DSM-V, include disturbance of attention, awareness, memory deficit, rapid onset, and fluctuations of the symptoms, in addition to psychomotor deficits, sleep disturbance, or sleep/awake cycle disturbance [46]. It should be mentioned that both DSM-V and ICD-10 do not provide clear methodology regarding the evaluation of attention and awareness or the determination of pre-existing neurocognitive disorders [47]; over the last four decades, the detection of delirium has remained stable [5]. However, it has been reported that less strict DSM-V criteria regarding attention and orientation make the diagnosis of delirium more inclusive [48]. Furthermore, ICD-10 and DSM-V represent the best expert consensus and available evidence, and their application is encouraged in both clinical and research contexts [49].

During the recent years, different screening tools for the detection of delirium have been developed. Here, we report the most frequent screening tools in the context of NHRs.

CAM is a widely used standardized delirium instrument for clinical and research purposes, designed to allow nonpsychiatric healthcare providers to detect delirium accurately [50,51].

CAM is based on DSM-III revised criteria and includes the following delirium features:(a)Fluctuating course,(b)Inattention,(c)Disorganized thinking,(d)Altered level of consciousness.

A further algorithm is applied for the classification of three delirium levels:(1)Subsyndromal delirium with only one of the four features present,(2)Subsyndromal delirium with two of the four features are present,(3)Full delirium: features a and b are present, along with features c and/or d [50].

At least 12 studies [11,15,17,18,25,27,28,39,40,41,42,43] in the context of nursing homes applied CAM, from 2003 to 2022 [52]. It should be mentioned that the prevalence of delirium still ranged widely, but CAM was helpful in the prediction of outcome (Table 1). This instrument has been validated from different studies applying DSM-III to IV, ICD-10, or consensus diagnosis including psychiatrist, geriatricians, and advanced practice nurses, with an overall sensitivity of 82% ranging from 69% to 91%, and a specificity of 99% ranging from 87% to 100% [52]. Of note, in the nursing home setting, the validity of CAM remains understudied.

NH-CAM is based on the four CAM features, which are modified using variables and items from the Minimum Dataset (MDS) Resident Assessment Protocol (RAP). NH-CAM includes the following features [36]:Mental function varies over the course of the day or mood decline over the last 90 days;Being easily distracted;Periods of altered perception or awareness of surrounding or episodes of disorganized speech or cognitive decline over the last 90 days;Periods of restlessness or periods of lethargy or behavior decline over the last 90 days.

Subsyndromal delirium level 1 is defined if any of the features is present, level 2 if any two of the four features are present, and full delirium if features 1 and 2 are present together with features 3 or 4.

Dosa et al., in a large cohort of 35,721, concluded that NH-CAM is a useful tool in estimating delirium related prognosis among NHR [36]. Other studies estimated the prevalence, incidence, and risk factors of delirium using NH-CAM or an adapted version to their population [22,26]. However, a clinical validation of NH-CAM is still lacking and future studies should evaluate if application of NH-CAM may result in the improvement of care and outcome in NHR.

NEECHAM is a nine-item instrument based on daily nursing practice, which evaluates the level of processing information, the level of behavior, and the physiological condition [53]. Compared to DSM-IV, NEECHAM covers 13 of the criteria for delirium with a total score that ranges from 0 to 30. A score of 24 or less indicates possible delirium. It has been demonstrated that NEECHAM scale is reliable for the detection of delirium by nurses in long-term care [54,55]. Regular daily administration may be difficult or burdensome; however, NEECHAM may be administrated when an acute change in mental status is present, as well as for a more in-depth assessment of delirium.

The 4AT is a brief screening tool including four items:Alertness,Abbreviated Mental Test-4,Attention,Acute change or fluctuating course.

Its score ranges from 0 to 12 points, where a score ≥ 4 suggests possible delirium. The 4AT tool is fast and does not require specific training for the implementation. It was initially validated and used in hospital settings and [56], and further large studies applied the 4AT tool in NHR settings [29,30]. A point prevalence analysis revealed that almost one to three persons in nursing homes are affected by delirium [30], and the 4AT score was not associated with the use of indwelling urinary catheters in NHRs [29]. It should be mentioned that a score of 4 points may be present in chronic cognitive impairment and cases where delirium is superimposed on dementia cannot be excluded. However, the adverse consequences of delirium non-detection are greater than delirium overdiagnosis.

The Delirium Diagnostic Tool-Provisional (DDT-Pro) is a brief scale designed to allow accurate delirium diagnosis by evaluating vigilance, comprehension, and sleep/awake cycle. The scores range from 0 to 9 points, and the recommended cutoff for delirium is a score ≤ 6. It has been reported that DDT-Pro sensitivity ranges from 88% to 100% and specificity ranges from 85.3% to 94.4% [57]. Data from a recent study reported that DDT-Pro is a valid tool to detect delirium when used by skilled nurses showing 77.2% sensitivity and 84% specificity; for a cutoff score ≤ 7, DDT-Pro sensitivity increased to 84.8% [37].

### 3.5. Prevention Strategies for Delirium in Nursing Home Settings

Dehydration was significantly associated with delirium onset [15,20,25], and persons in nursing homes may be highly vulnerable to dehydration due to the presence of other factors such as swallowing difficulties, lack of thirst, cognitive impairment, use of restraints, or substandard care. However, a randomized study where weight-based intervention of hydration was performed did not show reduction in delirium incidence during 4 weeks follow-up [58]. It should be mentioned that hydration status was controlled by bioelectrical impedance analysis, and older people with cardiovascular disease and diuretic therapy should be frequently monitored. In addition, hydration is one among the multiple risk factors which influence delirium, and modification of only one factor may not be sufficient for significant changes.

Despite the multifactorial characteristics of delirium, a multicomponent 16 month enhanced educational package to support care home staff to address key delirium risk factors did not show effectiveness in nursing home settings [59].

Furthermore, another multicomponent intervention study, targeting delirium risk factors (e.g., cognition, immobility, hydration, and undernutrition) in acutely ill long-term nursing home residents did not show significant results in the prevention of delirium. Differences regarding cognitive impairment prevalence at baseline between intervention and usual care group, resistance to some interventions because of physical impairment and disability, and lack of a formal interdisciplinary team were reported as possible explanations of the results [60].

The Transfusion Requirements in Frail Elderly (TRIFE) study was a prospective, assessor-blinded, randomized controlled trial focused on the role of red blood cell transfusion strategies on physical recovery or survival in frail older people with hip fracture after surgery. A post hoc analysis of this study, including NHR with hip fracture, revealed that liberal transfusion (Hb levels ≥ 11.3 g/dL) compared to the restrictive transfusion strategy (Hb levels ≥ 9.7 g/dL) reduced the occurrence of delirium. Results from this study indicated that, in older people with hip fractures, maintaining hemoglobin level above 11.3 g/dL reduced the rate of delirium [61].

Application of a software which correlates medication effects with physical, functional, and cognitive decline to foster early recognition of potential adverse drug effects showed that newly admitted NHRs experienced a lower rate of potential delirium compared to usual care NHR [62]. Another study focused on the implementation of an educational program directed toward nursing home physicians in reducing inappropriate prescription and improving health outcomes revealed that this intervention improved the inappropriate prescription of drugs and the occurrence of delirium [63].

A recent randomized control trial reported that doll therapy was more effective in reducing agitation, aggressiveness, professional caregiver burden, and the incidence of delirium compared to standard treatment in people living with dementia [64].

Of interest, other nonpharmacological interventions such as music therapy in people living with dementia improved the psycho-behavioral profile [65], and bright-light therapy was effective in reducing daytime sleep in NHRs with dementia [66]. Future studies may consider exploring the application of these interventions as a preventive strategy regarding the occurrence of delirium in nursing home settings.

Table 2 summarizes the characteristics of studies that explored interventions and prevention strategies for delirium in NHR. 

## 4. Discussion

Delirium is a common condition in the healthcare system with important health and socioeconomic costs. The majority of delirium research has been focused on the hospital setting, and NHRs remain an understudied field. According to a previous review which analyzed data from 15 studies, the prevalence of delirium in long-term care was between 1.4% and 70% [7]. However, long-term care has a broad definition and does not include only NHRs. In addition, since then, other studies focused on delirium have been published. In this review, we initially summarized the evidence referring to the prevalence and incidence of delirium in NHR. Although we focused on NHR and reviewed data from a relevant number of studies, the prevalence of delirium in NHR still ranged widely from 1.4% to 70.3%, while the incidence ranged between 10% and 60%. Differences related to diagnostic criteria, screening tools, baseline population characteristics such as the prevalence of dementia, and acute events are possible explanations for the wide range of prevalence and incidence.

Recognition of delirium may be difficult when dementia is present because of the overlapping of clinical characteristics such as hypoactivity and fluctuations of symptoms. Moreover, delirium superimposed on dementia is common in aged populations, and the present diagnostic criteria and screening tools provide a suboptimal performance [67]. Dementia is mainly characterized by memory deficits and characteristics of delirium such as alternation of attention, language, motor function, and sleep/awake cycle may also be present in dementia. However, these alternations in dementia without delirium are less accentuated and are characterized by an insidious and progressive onset [68,69,70]. It should be mentioned that both ICD-10 and DSM-V do not provide specific indications when dementia coexists.

Age, dementia, depression, and restraints were risk factors mentioned from different studies conducted in NHRs [71]. From our review a variety of demographic and social risk factors, comorbidities, conditions related to aging and diseases, malnutrition, and drugs were identified as risk factors for delirium onset in NHR. Age, dementia, physical restraints, and falls are common and significant risk factors in nursing homes. Not all the studies conducted in NHRs reported age as a significant risk factor. The majority of studies in nursing homes included populations at advanced age, with a more homogeneous age distribution compared to other settings. In NHRs, the severity of dementia was measured using different scales such as the Mini-Mental State Examination or Montreal Cognitive Assessment Scale. However, dementia resulted as an independent risk factor in most of the NHR studies. This result is line with other studies in different settings [72]. The association of falls and restrains with delirium resulted significant after adjustment for confounders.

Furthermore, NHRs with COVID-19 infection experienced a higher prevalence rate of delirium ranging from 49.2% to 57.9% compared to hospitalized people aged over 65 years, where an overall prevalence of 28% was reported [73]. Clinical presentation of COVID-19 disease may be masked by delirium, and SARS-CoV-2 infection should be considered in people experiencing delirium. In addition, cases of delirium after COVID-19 vaccination have been reported. It should be mentioned that delirium resolved without complications [42]. However, healthcare providers should monitor for delirium after vaccination in this population.

Delirium experienced in NHR is associated with poor outcome regarding survival, rehospitalizations, and functional decline. In other settings, delirium experienced during hospitalization also increases likelihood of nursing home placement and is associated with about twofold increased mortality risk within 2 years [8,74]. In persons newly admitted into post-acute care facilities from acute care hospitals, delirium is associated with worse activities of daily living (ADL) and instrumental activities of daily living (IADL) recovery, indicating that delirium may have important consequences for functional performance and dependence [75]. Of note, the relationship between delirium and dementia seems bidirectional; delirium may be associated with long-term cognitive decline and dementia development [76].

CAM, NH-CAM, and NEECHAM are the most frequent screening tools applied in NHR. Despite their confirmed validity in different settings, the clinical validity of these tools in nursing homes is still understudied. Furthermore, routine use of delirium screening tools in nursing homes is not common [77]. It should be mentioned that screening tools may be time-consuming and not always easy to use; for these reasons, an “ideal” delirium screening tool in the context of NHR should be fast, easy to use, and able to be integrated in everyday work. Recently, new screening tools such as DDT-Pro, 4AT, and I-AGeD have shown good performance in nursing homes, and further validation studies should be performed with larger populations. In addition, the impact of these tools in the prognosis and management of NHR should be evaluated in-depth.

Previous studies have demonstrated that interventions for delirium prevention are highly effective in decreasing occurrence of delirium, with more than 50% odds reduction, and approximately one million cases of delirium in the hospital settings could be preventable by multicomponent nonpharmacologic interventions [78]. However, limited evidence has been identified on interventions for preventing delirium in a long-term care setting [79]. Hydration, multicomponent interventions, and educational strategies neither reduced the incidence of delirium nor influenced the outcome [58,59,60]. Application of software and educational programs which are helpful in the identification of possible adverse drug effects seems promising for the reduction in delirium incidence [62,63]. In older people with specific conditions such as hip fracture, transfusion resulted in a reduction in delirium rate [61]. Furthermore, other interventions focused on psychologic behaviors such as doll, bright-light, and music therapy should be further investigated regarding their role in reducing the incidence of delirium.

The identification and the correction of life-threatening causal factors and the underlying causes of delirium are crucial once delirium has been detected. Adaptation of nonpharmacological interventions such as avoiding unnecessary catheters and physical restraints, approaching people at the bedside, and tools to facilitate orientation may be useful to improve mild-to-moderate agitation. It is also recommended to avoid poor communication and inadequate staff attention to promote adequate nutrition and fluid intake. Exclusion of pain, constipation, and urinary retention should be considered as important causes of agitation in older people. Several guidelines offer practical recommendations regarding delirium management, as well as indications regarding antipsychotic use [80,81].

This study had some limitations; we provided a review of the published studies and not a systematic review. It is possible that other studies could provide important information regarding delirium epidemiology, risk factors, outcome, and prevention in NHR. Furthermore, being a narrative review, the quality of studies was not evaluated.

## 5. Conclusions

The prevalence and the incidence of delirium in nursing homes are frequent but range widely. Different diagnostic criteria, a variety of detection tools, and delirium superimposed on dementia may in part explain the wide range of delirium occurrence in NHR. A considerable number of risk factors associated with delirium onset have been described including demographic, social, and nutritional factors, as well as comorbidities and therapy. In addition to well-known risk factors such as age and dementia, physical restraints and falls also need to be considered as potential risk factors in nursing home settings. SARS-CoV-2 infection should be considered in NHRs with delirium, since delirium may mask COVID-19 disease. Different delirium detection tools have been developed; however, the clinical validity of most of them is still understudied in nursing homes. Prevention strategies such as software interventions to monitor therapy, educational programs applied among healthcare providers, delirium routine screening by nurse staff, and other interventions regarding anemia, behavioral, and psychological symptoms give promising results; however, further studies should investigate their effectiveness in reducing delirium incidence in NHR.

## Figures and Tables

**Table 1 healthcare-10-01544-t001:** Main characteristics of the studies focused on delirium in nursing home residents.

First Author and Year	Study Design	Total Population	Mean Age and Female Prevalence	Delirium Prevalence and or Incidence	Delirium Diagnostic Criteria and Instrumental Tool	Risk Factors	Main Outcome	Main Results
**Sandberg et al., 1998 [12]**	Observational	202	84.2 years F: 65.3%	Prevalence: 58%	DSM-III-R	N/A	N/A	Delirium is frequent in different types of care settings.
**Mentes et al., 1999 [13]**	Observational	2318	83 years F: 76.6%	Prevalence: 13.98%	MDS	Inadequate fluid intake OR = 3.4, 95% CI (2.99–3.81); Falls OR = 1.61, 95% CI (1.26–1.96); Dementia OR = 1.44, 95% CI (1.21–1.68)	N/A	Delirium prevalence was lower compared to other studies.
**Laurila et al., 2003 [16]**	Cross-sectional	195	Mean age N/A F: 90.8%	Prevalence: 10.1–24.9%	DSM-III DSM-III-R DSM-IV ICD-10	N/A	N/A	DSM-IV simplified recogntition of delirium.
**Dosa et al., 2007 [36]**	Retrospective cohort	35,721	Mean age N/A F: 73.3%	Prevalence: 1.4%	MDS NH-CAM	N/A	Mortality risk for Delirium OR: 1.96, 95% CI (1.71–2.26).	NH-CAM succesfully predicted outcome.
**Culp and Cacchione 2008 [17]**	Observational	312	88.5 years F: 76.6%	Prevalence: 21.8%	CAM NEECHAM MMSE Vigilance A error score	Increased % FFM Increased % BCM	N/A	NHRs often present undernutrition or become undernourished.
**Voyer et al., 2009 [18]**	Cross-sectional	155	86.3 years F: 73.6%	Prevalence: 29–70.3%	DSM-III DSM-III-R DSM-IV CAM	N/A	N/A	Prevalence for delirium depends on diagnostic tool used.
**Voyer et al., 2009 [19]**	Cross-sectional secondary analysis	155	86.3 years F: 73.6%	Prevalence: 70.3%	CAM	Age OR = 1.07, 95% CI (1.05–1.10); Severity of dementia OR = 1.05, 95% CI (1.03–1.07)	N/A	Age and dementia are the most important risk factors associated with delirium.
**Yang et al., 2009 [21]**	Cross-sectional	441	84.1 years F:64.6%	Prevalence: 68.9%	MDS	N/A	No dementia: hypo-mild class delirium HR = 1.99, 95%CI (1.02–3.86) Mixed class: HR = 1.98, 95% CI (1.08–3.64) Dementia: hypo-mild class delirium: HR = 3.98, 95% CI (1.76–8.98)	Delirium severity and psychomotor features provide important prognostic information.
**Voyer et al., 2010 [20]**	Cross-sectional Secondary analysis	155	86.3 years F: 73.6%	Prevalence: 70.3%	CAM	Multiple risk factors.	N/A	Multiple factors play a key role in delirium onset in people living with dementia.
**von Gunten and Mosimann 2010 [22]**	Observational	11,745	N/A	Prevalence: 6.5%	NH-CAM	Increased dependence; Dementia; Psychiatric diseases.	N/A	Subsyndromal and full delirium are common in NHR upon admission.
**McCusker et al., 2011 [11]**	Prospective	279	Mean age N/A F: 56.3%	Prevalence: Cohort A (MMSE ≥10): 3.4% (MMSE <10): 33% Incidence: Cohort A: 1.6 per 100 persons per week Cohort B: 1.7 per 100 persons per week	CAM	Dementia OR = 5.85, 95% CI (1.12–30.53); Severe ADL impairment OR = 8.39, 95% CI (1.42–49.75); Depressive symptoms OR = 3.43, 95% CI (1.09–10.8)	N/A	Delirium is an important clinical problem in people with moderate to severe cognitive impairment.
**Arinzon et al., 2011 [25]**	Prospective	322	79.86 years F: 74%	Prevalence: 34 %	CAM DSM-III-R	Infections DehydrationElectrolyte imbalance Drugs toxicity Drugs withdrawal	Infections, malnutrition, advanced age, and chronic heart failure.	Recognition, identification, correction of delirium, and risk factors may influence the outcome.
**Ishii et al., 2011 [24]**	Cross-sectional	2330	N/A	Prevalence: 14.3%	CAM	N/A	N/A	Delirium, delusions, depression, and severe pain were associated with rejection of care.
**McCusker et al., 2011 [23]**	Prospective, secondary analysis	235	Mean age N/A F: 57.8%	Prevalence: 33.3%	CAM	N/A	Delirium or decline in cognitive and functional status predict mortality HR = 1.77, 95% CI (1.13–2.28)	Delirium symptoms observed by nurses improve the detection of delirium and prediction of outcomes.
**Boorsma et al., 2012 [26]**	Cohort study	828	Mean age N/A F: 67%	Prevalence: 8.9% Incidence: 20.7 per 100 person-years	NH-CAM	Dementia OD = 3.1, 95% CI (2.0–5.0); Parkinson’s disease OD = 2.4, 95% CI (1.0–5.9); Chair restraints OD = 2.4, 95% CI (1.3–4.4)	N/A	Focus on use of restraints in nursing homes may help to prevent delirium.
**Boockvar et al., 2013 [39]**	Prospective observational cohort study		76.2 years F: 40%	Incidence: 17.7%	CAM	Urinary tract infections Respiratory infections Female gender	Cognitive decline OR: 4.59, 95% CI (1.99–10.59)	Delirium occurred frequently as a complication of acute illness, and was associated with cognitive function decline.
**Kosar et al., 2017 [27]**	Retrospective cohort study	5,346,581	81.2 years F: 63%	Prevalence: 4.3%	CAM	Dementia Male gender Comorbidities Visual and auditory functional impairment	Mortality 1 year RR = 1.54, 95% CI (1.53–1.54); Rehospitalization RR = 1.42, 95% CI (1.40–1.43); Functional improvement RR = 0.83, 95% CI (0.82–0.83).	Early identification of delirium may improve outcome.
**Perez-Ros et al., 2018 [28]**	Retrospective cross-sectional case–control study	316	N/A	Prevalence: 60.7%	DSM-IV CAM	Infections OR = 7.08, 95% CI (3.30–15.02); Dementia OR = 3.14, 95% CI (1.81–5.45); Anticholinergic drugs OR = 2.98, 95% CI (1.34–6.60); Depression OR = 1.92, 95% CI (1.03–3.56); Urinary incontinence OR = 1.73, 95% CI (0.97–3.08)	N/A	Infections, dementia, anticholinergic drugs, depression, and urinary incontinence are predictive for delirium.
**Bo et al., 2018 [29]**	Multicenter observational study	1454	84.4 years F: 70%	Prevalence: 36.7%	4AT	N/A	N/A	No significant association between IUC and delirium in NHRs.
**Cheung et al., 2018 [40]**	Retrospective cohort study	1571	83.3 years F: 66.2%	Incidence: 40.4%	CAM	Dementia OR = 2.54, 95% CI (1.99–3.25); pain OR = 1.64, 95% CI (1.25–2.16); antipsychotics OR = 1.88, 95% CI (1.40–2.51)	N/A	Dementia, pain, and antipsychotics were associated with the onset of delirium.
**Morichi et al., 2019 [30]**	Multicenter observational	1454	84.4 years F: 70.2 %	Prevalence: 36.8%	4AT	Education OR = 0.94, 95% CI (0.91–0.97); Dementia (OR = 3.12, 95% CI (2.38–4.09); Functional dependence OR = 6.13, 95% CI (3.08–12.19); Malnutrition OR= 4.87, 95% CI (2.68–8.84); Antipsychotics OR = 2.40, 95% CI (1.81–3.18); Physical restraints OR = 2.48, 95% CI (1.71–3.59)	N/A	Delirium is common in NHRs. Simple assessment tools might facilitate its recognition in this population.
**Perez-Ros et al., 2019 [41]**	Cohort trial nested case–control study	443	85.7 years F: 78.3%	Incidence: 18.7%	DSM-IV CAM	Dementia OR = 2.74, 95% CI (1.49–5.04); Falls OR = 2.45, 95% CI (1.49–3.69); Neuroleptics OR = 2.39, 95% CI (1.23–4.65); Anticholinergics OR = 1.87, 95% CI (0.95–3.69).	N/A	Dementia is a predisposing factor. Falls and neuroleptic drugs are predictive factors.
**Sepulveda et al., 2019 [44]**	Cross-sectional prospective study	131	76.3 years F: 51.1%	Incidence: 22.1%	DSM-V	Age OR = 1.076, 95% CI (1.006–1.151); Widowhood OR = 3.701, 95% CI (1.133–12.091); Physical restraints OR = 9.221, 95% CI (2.394–35.521); Intravenous catheter OR = 46.019, 95% CI (4.953–427.579); Hearing impairment OR = 4.255, 95% CI (1.353–13.379)	N/A	Underdiagnosis of delirium in nursing homes. Dementia was not a risk factor.
**Shi et al., 2020 [38]**	Retrospective cohort study	146	85 years F: 55.9%	Prevalence: 57.3%	MDS	N/A	N/A	Predictors of COVID-19 infection: male sex, bowel incontinence, and staff.
**Sepulveda et al., 2021 [37]**	Prospective cross-sectional	262	77.1 years F: 43.9%	Prevalence: 30.1%	DDT-Pro 4AT	N/A	N/A	DDT-Pro is valid to detect delirium in nursing setting.
**Sabbe et al., 2021 [31]**	Multicenter cross-sectional	338	84.7 years F: 67.5%	Prevalence: 15%	DOSS	Falls OR = 2.76, 95% CI (1.24–6.14); MoCA Score OR = 0.69, 95% CI (0.63–0.77)	N/A	Delirium affects almost 15% of NHRs.
**Beiting et al., 2021 [33]**	Single-center, retrospective observational cohort study	122	N/A	Prevalence: 49.2%	N/R	N/A	N/A	Predominant symptoms for COVID-19 were low-grade fever, anorexia, delirium, and fatigue.
**Urfer Dettwiler et al., 2022 [34]**	Cross-sectional prospective single-center pilot study	85	85.5 years F: 64.7 %	Prevalence: 5.9%	DSM-V I-AGeD mCAM-ED	N/A	N/A	I-AGeD showed a sensitivity of 60% and specificity of 94% at a cutoff point of ≥4 to indicate delirium.
**Mak et al., 2022 [42]**	Case series	40	82 years F: 55%	Incidence: 10%	DSM-V CAM	N/A	N/A	Delirium after COVID-19 vaccination resolved without complications, in contrast with complications of COVID-19 infection itself.
**Skretteberg et al., 2022 [43]**	Prospective cohort	145	84.2 years F: 69.7%	Incidence: 60%	CAM	Infections Benzodiazepines Vascular dementia	N/A	Infection was the most common cause of delirium.
**Zucchelli et al., 2022 [35]**	Observational	501	N/A	Prevalence: 31.3%	4AT	Age: OR = 1.01 95% CI (1–1.03); Dementia: OR = 9.31 95% CI (7.08–12.34); Calf circumference OR = 0.94 95% CI (0.92–0.97)	N/A	Calf circumference is independently associated with delirium.

FFM: fat-free mass; BCM: body cell mass; MoCA: Montreal Cognitive Assessment; IUC: indwelling urinary catheter; OR: odds ratio; CI: confidence interval.

**Table 2 healthcare-10-01544-t002:** Main Characteristics of studies investigating prevention strategies and interventions to reduce delirium occurrence in nursing homes.

First Author and Year	Study Design	Population	Diagnostic Criteria	Intervention	Follow-Up	Main Results
**Culp et al., 2003 [58]**	Randomized study	Treatment group: 53 Control group: 45	NEECHAM	Individual oral fluid intake	4 weeks	No delirium incidence reduction: RR = 0.84, 95% CI (0.18–4.0)
**Lapane et al., 2011 [62]**	Randomized study	Baseline: Intervention group: 1711 (491 interventions) Usual care: 1492 (baseline); Intervention period: Intervention group: 1769 Usual care: 1769	NH-CAM	Multicomponent educational intervention based on Geriatric Risk Assessment MedGuide (GRAM) software	Monthly assessment, minimum 12 months	In home: Deaths: HR = 0.89, 95% CI (0.73–1.08); Hospitalizations: HR= 1.11, 95% CI (0.94–1.31); Hospitalizations for ADE: HR = 1.12, 95% CI (0.63–1.98); Falls: HR = 0.94, 95% CI (0.85–1.04); Potential delirium indicator: HR = 0.93, 95% CI (0.8–1.09); New admits: Deaths: HR = 0.88, 95% CI (0.66–1.16) Hospitalizations: HR= 0.89, 95% CI (0.72–1.09); Hospitalizations for ADE: HR = 1.47. 95% CI (0.58–3.71); Falls: HR = 1.03, 95% CI (0.92–1.15) Potential delirium indicator: HR = 0.42, 95% CI (0.35–0.52)
**Garcia-Gollarte et al., 2014 [63]**	Prospective randomized multicenter study	Intervention group: 372 Control group: 344	N/R	Educational interventions on drug use regarding geriatric population in physicians working in NHR	3 months	Delirium increased in the control group (from 0.04 to 0.14 per resident, *p* = 001) but was reduced significantly in the intervention group (0.08 at baseline, 0.03 at the end of the study *p* = 0.035)
**Siddiqi et al., 2016 [59]**	Cluster randomized feasibility study	Intervention group: 75 Control group: 85	CAM	Multifaceted enhanced educational package	16 months	Prevalence of delirium: RR = 0.57, 95% CI (0.15–2.19); Incidence of delirium RR = 0.62, 95% CI (0.16–3.39); Hospitalizations HR = 0.72, 95% CI (0.38–1.36); Deaths HR 0.72, 95% CI (0.26–2.11)
**Blandfort et al., 2017 [61]**	Post hoc analysis of a randomized control trial	Intervention group: 90 Control group: 89	CAM	TRIFE trial: RBC transfusion strategy (Hb: 9.7 g/dL or the liberal strategy (Hb: 11.3 g/dL) NHR after hip fracture surgery	10 days after transfusion	Delirium occurrence: RR = 0.46, 95% CI (0.22–0.97); MMSE score: RR = 1.22, 95% CI (1.08–1.36); Mortality: RR = 0.39, 95% CI (0.12–1.2)
**Boockvar et al., 2020 [60]**	Cluster randomized controlled trial	Intervention group:114 Control group: 105	CAM	Multicomponent intervention designed to ameliorate delirium risk factors: cognitive impairment, immobility, dehydration, and malnutrition	1 month	Delirium incidence: RR 1.14, 95% CI (0.78–1.6); Cognitive performance MMSE score: mean 2.1 SE: 0.2 vs. 1.5 SE: 0.1, *p* = 0.037
**Santagata et al., 2021 [64]**	Randomized control trial	Intervention group: 26 Control group: 26	CAM	Empathy dolls were administered two times a day for 2 h in the morning, 2 h in the afternoon, and in case of agitation, aggressiveness, and or wandering	90 days	Reduced incidence of delirium (*p* = 0.0211); Reduced agitation, aggressiveness, and dysphoria (*p* < 0.01); Reduced caregiver burden *p* < 0.0001); No reduction in antipsychotic administration

N/R: not reported; RBC: red blood cell; NHR: nursing home resident; RR: relative risk; HR: hazard ratio; CI: confidence interval.

## Data Availability

Not applicable.

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
