# Peer review of "Delirium in Nursing Home Residents: A Narrative Review"

_healthcare, 2022, doi:10.3390/healthcare10081544_

Round 1
Reviewer 1 Report
The article aims to review an understudied field, delirium in nursing homes (NH). I think it is a very important and interesting field, in which a review may be very useful for further research.
I have, however, some concerns about your article:
1. The work refers to other previous reviews but does not explain what is new in relation to them. It also duplicates some results by discussing an article both individually and within the context of an earlier review.
2. Methodology should be more detailed. How many reviewers participated and what were their roles? What was the search strategy, keywords used, etc.? Did you use any criteria for determining quality of the articles assessed?
3. You should consider separate sections for results and discussion, so that we may differentiate the actual review from your interpretation of it.
4. The Introduction section is not well focused. It is mainly based on results of studies in settings different from NH, which lack a clear connection with the goals of your review. It could be useful, for example, to emphasize how delirium studies in other fields find results for which evidence is lacking in NH, in order to justify your focus on the latter. Also, you should revise if sentences in other sections would fit better as part of the introduction, e.g. in lines 142 to 145: “Delirium results from a combination…”
5. Sections 2 to 6, which currently read like a list of references, should be rewritten to include, at some point, a summary of relevant results. Related to this point, Table 1 might be useful to summarize all articles (currently, not all articles are included and some of the highlighted results are not to be found elsewhere in your review).
In addition, there are some references that need to be clarified:
6. The first part of section “2. Delirium Screening Tools” says “The diagnosis of delirium is guided…” by DSM-5 criteria. Maybe you want to say “… should be guided…” but it could be important to say why these criteria and no others, (you do not mention for example ICD-10 or ICD-11).
7. When you say “the screening tools used in the NHR”, what do you mean? The most frequently used? The scales validated in this population?
8. When talking about CAM, you do not focus on NHs. Also, the article you reference when discussing CAM sensitivity is outdated: newer meta-analysis show that CAM tends to have lower sensitivity values. The description of the NH-CAM items is not accurate, they are based on the original items but adapted to be extracted from de Minimum Data Set (MDS) Resident Assessment Protocol (RAP).
9. Some studies are referenced without giving relevant information, e.g. when talking about 4AT (lines 98 and 99) you say: “and further large studies applied 4AT tool NHR settings (25, 26)” without specifying what characteristics and which results of these studies are relevant to NHRs. In general, you describe the main characteristics of the studies, but do not give any further information, for example comparing them to other studies, pointing out what is most relevant about them in NHR settings etc.
10. Some statements are given without supportive evidence e.g. in line 126 “Furthermore, NHR affected by COVID-19 disease have a high incidence of delirium”, the question would be, how high?
11. In describing the DDT-Pro, you say that it evaluates “cognitive performance, vigilance and circadian sleep-awake cycle”, whereas it should be: “vigilance, comprehension and sleep-wake cycle”.
12. If using literal quotes, you should make sure that the wording is the same as in the original article i.e. in lines 109-11: “The prevalence was higher in population affected by dementia, while lower frequency was found when full delirium criteria were used or severe dementia was considered and exclusion criteria”.
13. In lines 118-119 you say: “In the above-mentioned studies AT4 was performed by physicians and DOSS by nurses”, what is the source of this statement?
Author Response
Reviewer #1
The article aims to review an understudied field, delirium in nursing homes (NH). I think it is a very important and interesting field, in which a review may be very useful for further research.
I have, however, some concerns about your article:
- The work refers to other previous reviews but does not explain what is new in relation to them. It also duplicates some results by discussing an article both individually and within the context of an earlier review.
REPLY: Thank You for your comments. In the revised version of our manuscript we underlined the novelty of our review: a) compared to previous important reviews which included in general long-term settings we focused on nursing homes and b) and provided a review of studies reporting data not only on epidemiology and risk factors but also included outcomes and interventions. Furthermore, we discussed the most frequent delirium screening tools used in nursing homes, reporting their utility, validity and suggested new direction for future studies. Please check the revised version of our manuscript page 22 line 645-655; line 669-678; page 23 line 750-753, page 24 lines: 772-780; and page 24 lines 838-851. In the revised version of our manuscript, we provided a result section where original research data focused on NHR were discussed. Also, table 1 was modified and only original research articles were added. Multiple studies reporting data from the same cohort were considered, if important information regarding other aims of our review was included.
- Methodology should be more detailed. How many reviewers participated and what were their roles? What was the search strategy, keywords used, etc.? Did you use any criteria for determining quality of the articles assessed?
REPLY: Thank You for the suggestions. We added the Methods paragraph and provided details regarding our search strategy: search terms, reviewers and their role, the time period and language. Please find in the revised version of our manuscript these modifications in page 2 lines 108-121.The Reviewer pointed an important question regarding the quality of the articles included in this review. We did not follow specific quality criteria assessments for the included articles, since we did not perform a systematic review study. However, we added this point as a limitation of our review. Please check page 24 limitations paragraph, lines 833-836.
- You should consider separate sections for results and discussion, so that we may differentiate the actual review from your interpretation of it.
REPLY: Following the reviewer's comment we differentiate our article in separate sections for results and discussion.
- The Introduction section is not well focused. It is mainly based on results of studies in settings different from NH, which lack a clear connection with the goals of your review. It could be useful, for example, to emphasize how delirium studies in other fields find results for which evidence is lacking in NH, in order to justify your focus on the latter. Also, you should revise if sentences in other sections would fit better as part of the introduction, e.g. in lines 142 to 145: “Delirium results from a combination…”
REPLY: Thank you for the comment and suggestions. We modified the introduction underling that the present evidence regarding delirium (epidemiology, outcome etc) is focused mainly in hospital, palliative care or general long-term care setting and despite the characteristics of nursing home residence which may make them prone to delirium development, the evidence in NHR is lacking. The sentence that the Reviewer indicated is part of the introduction in the revised version of the manuscript. Please check page 1 lines: 25-31;40-41.
- Sections 2 to 6, which currently read like a list of references, should be rewritten to include, at some point, a summary of relevant results. Related to this point, Table 1 might be useful to summarize all articles (currently, not all articles are included and some of the highlighted results are not to be found elsewhere in your review).
REPLY:According to Reviewers suggestion all these sections were rewritten. We provided separate points for review's results and table 1 was modified including all relevant articles within the scope of our review. In table 1 we summarized all articles with relevant data regarding prevalence, incidence and also risk factors and outcome. We provided and additional table, table 2 where we summarized all articles with relevant data regarding preventive measures and interventions to reduce delirium occurrence in nursing homes.
In addition, there are some references that need to be clarified:
- The first part of section “2. Delirium Screening Tools” says “The diagnosis of delirium is guided…” by DSM-5 criteria. Maybe you want to say “… should be guided…” but it could be important to say why these criteria and no others, (you do not mention for example ICD-10 or ICD-11).
REPLY: Thank You for the suggestions. The Reviewer is right, and we modified the text reporting should be guided. We introduced information regarding ICD-10. And also provided comments why diagnosis of delirium should be guided by DSM/ICD criteria and some limitations of these criteria. Please check page 3 lines 209-211; page 4: 222-231.
- When you say “the screening tools used in the NHR”, what do you mean? The most frequently used? The scales validated in this population?
REPLY: We summarized the most frequent screening tools in nursing homes, which resulted from our review. Surprisingly, most of these tools are validated but not in nursing homes settings. Some studies that we mentioned in the section 3.1 and 3.4 have performed validation studies for few tools in nursing homes (for instance I-AGeD, DDT-pro). We pointed this important detail in the revised version of our manuscript. Please check: section 3.4 pages 4 and 5, and discussion section, page 22 lines 771-793.
- When talking about CAM, you do not focus on NHs. Also, the article you reference when discussing CAM sensitivity is outdated: newer meta-analysis show that CAM tends to have lower sensitivity values. The description of the NH-CAM items is not accurate, they are based on the original items but adapted to be extracted from de Minimum Data Set (MDS) Resident Assessment Protocol (RAP).
REPLY: In the revised version of our manuscript we focused on the studies conducted in nursing homes where CAM was applied, and referred to a newer systematic review and meta-analysis study regarding CAM sensitivity. NH-CAM description was corrected. Please check page 4 lines 248-254 and lines 256-267.
- 9. Some studies are referenced without giving relevant information, e.g. when talking about 4AT (lines 98 and 99) you say: “and further large studies applied 4AT tool NHR settings (25, 26)” without specifying what characteristics and which results of these studies are relevant to NHRs. In general, you describe the main characteristics of the studies, but do not give any further information, for example comparing them to other studies, pointing out what is most relevant about them in NHR settings etc.
REPLY: We added characteristics of the studies with relevant information regarding application of 4AT in nursing homes. In briefly the most important information is relate to dementia-delirium overlapping. Please check page 5 lines 355-359.
- Some statements are given without supportive evidence e.g. in line 126 “Furthermore, NHR affected by COVID-19 disease have a high incidence of delirium”, the question would be, how high?
REPLY: We apologize for the wrong phrase, in this part the correct phrase is:..''NHR with COVID-19 infection experience a higher prevalence rate of delirium 49.2-57.9% compared to hospitalized patients aged over 65 years where an overall prevalence of 28% has been reported'' ..We referred to the meta-analysis by She et al 2021 (reference number 71 in the revised version of our manuscript). Please check page 23 lines 750-752.
- In describing the DDT-Pro, you say that it evaluates “cognitive performance, vigilance and circadian sleep-awake cycle”, whereas it should be: “vigilance, comprehension and sleep-wake cycle”.
REPLY:Thank you! This was corrected as indicated by the Reviewer. Please check page 5 line 361.
- If using literal quotes, you should make sure that the wording is the same as in the original article i.e. in lines 109-11: “The prevalence was higher in population affected by dementia, while lower frequency was found when full delirium criteria were used or severe dementia was considered and exclusion criteria”.
REPLY: We referred to the review by de Lange et al 2011, and wanted to summarize the reasons for the high heterogeneity, however, in the revision version of our manuscript this part was cancelled to avoid confusion.
- In lines 118-119 you say: “In the above-mentioned studies AT4 was performed by physicians and DOSS by nurses”, what is the source of this statement?
REPLY: Thank you for the precise revision. In the article by Sabbe et al 2021, in procedure section is reported that DOSS was performed by research nurses, while Morrichi et al 2018 reported:. untrained physicians or nurses. Anyhow, for not creating confusion also this phrase was cancelled.
We wish to thank the Reviewer for His/Her constructive criticism which helped us to improve our manuscript.

Reviewer 2 Report
This paper is a narrative review of delirium in nursing home residents. Subjects covered include the incidence / prevalence of delirium in this population, delirium screening tools, risk factors, outcome, and prevention strategies.
The topic is of considerable general interest given the demographic shift occurring all around the world, and the material presented - though not new - is well-organized and can serve as a useful guide for students, healthcare workers and researchers.
The current version of this paper can be improved by corrections or clarifications in the following areas:
1. As this is a narrative review, a formal flow diagram / PRISMA checklist is not required. However, it would still be useful to provide some additional details about the search strategy adopted (search terms, time period covered, language(s) included) in section 1 of the paper.
2. It would be more logical to place the section on "Delirium screening tools" after the sections on "incidence and prevalence of delirium", "risk factors" and "adverse outcomes", as these sections provide the necessary background to understand the indications for the use of such screening tools.
3. In the section on "risk factors", it would be helpful to briefly summarize the most common aetiologies of delirium in this population (some of this information is already provided in the Table, but it can be summarized in 1-2 sentences in the text as well.)
4. Given the population being discussed here, a brief discussion of the links between dementia and delirium (and the clinical phenomenon of delirium superimposed on dementia) should be included in this paper - ideally as a separate section, but if this is not possible, it can be assigned a separate paragraph in the section on "risk factors".
5. The authors have provided a citation to a single study suggesting that doll therapy may reduce the incidence of delirium in patients with dementia. What other interventions are of use in reducing or preventing delirium in this population? (As this is a general review, it is preferable to discuss all available, evidence-based approaches rather than to focus on a single one.)
6. A brief discussion of the management of delirium in a nursing home setting (especially as some of these patients will not be referred to or admitted in hospitals) should be provided near the end of the manuscript. (If there is an existing review or guideline which already addresses the management of delirium in this population, the authors can simply state "For a discussion of the management of delirium in this setting, see the review / guideline by X et al. [ref no.]").
Author Response
Reviewer #2
This paper is a narrative review of delirium in nursing home residents. Subjects covered include the incidence / prevalence of delirium in this population, delirium screening tools, risk factors, outcome, and prevention strategies.
The topic is of considerable general interest given the demographic shift occurring all around the world, and the material presented - though not new - is well-organized and can serve as a useful guide for students, healthcare workers and researchers.
The current version of this paper can be improved by corrections or clarifications in the following areas:
- As this is a narrative review, a formal flow diagram / PRISMA checklist is not required. However, it would still be useful to provide some additional details about the search strategy adopted (search terms, time period covered, language(s) included) in section 1 of the paper.
REPLY: Thank you very much for your comments. Following the Reviewer's suggestions, we provided details regarding our search strategy: search terms, reviewers, the time period and language. Please check in the revised version of our manuscript in Methods paragraph, page 2, lines 108-121.
- It would be more logical to place the section on "Delirium screening tools" after the sections on "incidence and prevalence of delirium", "risk factors" and "adverse outcomes", as these sections provide the necessary background to understand the indications for the use of such screening tools.
REPLY: According to the Reviewer suggestion, we re-organized our manuscript and placed delirium screening tools after prevalence and incidence, risk factors and outcome. Please check pages :2-4.
- In the section on "risk factors", it would be helpful to briefly summarize the most common aetiologies of delirium in this population (some of this information is already provided in the Table, but it can be summarized in 1-2 sentences in the text as well.)
REPLY: We rephrased the ''risk factors'' section and summarized risk factors for delirium in nursing homes following an order which corresponds to: demographic, social, comorbidities, and drugs. Age, dementia, physical restraints and falls result the most common triggers for delirium. This sentence was added in discussion section, page 22, lines 672- 674, and please also check conclusions: page 24 lines: 841-844.
- Given the population being discussed here, a brief discussion of the links between dementia and delirium (and the clinical phenomenon of delirium superimposed on dementia) should be included in this paper - ideally as a separate section, but if this is not possible, it can be assigned a separate paragraph in the section on "risk factors".
REPLY:Thank you for your suggestion. This is a very important topic for dementia in nursing homes, which influences the recognition of delirium. We added a paragraph in discussion section page 23 lines 665-671.
- The authors have provided a citation to a single study suggesting that doll therapy may reduce the incidence of delirium in patients with dementia. What other interventions are of use in reducing or preventing delirium in this population? (As this is a general review, it is preferable to discuss all available, evidence-based approaches rather than to focus on a single one.)
REPLY:We extended the section regarding prevention strategies for delirium in nursing homes. More detailed information was added regarding multicomponent interventions, educational programs, and other interventions regarding behavioral, and sleep-awake cycle. Furthermore, characteristics of studies investigating prevention strategies for delirium in nursing homes were summarized in table 2. Please check page 16 lines 459-503.
- A brief discussion of the management of delirium in a nursing home setting (especially as some of these patients will not be referred to or admitted in hospitals) should be provided near the end of the manuscript. (If there is an existing review or guideline which already addresses the management of delirium in this population, the authors can simply state "For a discussion of the management of delirium in this setting, see the review / guideline by X et al. [ref no.]").
REPLY:Thank you for the suggestion. We provided a paragraph regarding management of delirium and as the Reviewer indicated referred to the available guidelines. Please check page 23 lines 795-803.
We wish to thank the Reviewer for His/Her constructive criticism which helped us to improve our manuscript.

Reviewer 3 Report
Please find below some points about the first part of your article. Many of the points will be applicable in principle throughout the remainder of the article - e.g. consistency, language, clarifications.
Abstract, line 11. Two minor points about language. Firstly, when you talk about ‘elderly patients’ do you actually mean patients or elderly people more generally? It would be good if you can only use ‘patients’ when you’re talking about someone being treated in a healthcare setting and the context makes this clear. Secondly, instead of saying ‘2-5% of elderly world-wide’ it would be better to say ‘2-5% of elderly people world-wide’. (The second point also applies on page 1, line 45 where this phrase is also used)
Page 1, line 33. Similar point about the use of ‘patients’. When saying ‘a study including patients’ is it actually patients or people? I don’t know what the study was about , but it might be useful to give some context by saying something like ‘a study including hospital patients’ (or wherever the study was based) to help the reader know what it was about.
There are other places where the use of ‘patients’ might need to be checked or clarified with some context, but I’ll leave it up to you to go through these rather than pointing out every case.
Page 1, line 36. You don’t need both ‘of’ and ‘for’ when you say ‘risk of for mortality’. I would get rid of the ‘for’
Page 2, lines 49-52. The sentence ‘Nevertheless,…measures.’ doesn’t make sense to me. I suggest rewriting it, maybe to something like ‘Nevertheless, delirium often remains undetected and underdiagnosed. Many reasons, such as a lack of education and training for healthcare providers, poor implementation of validated detection tools, and staff attitudes, make it challenging to detect delirium and implement preventive and therapeutic measures.’
Page 2, lines 52-54. I would have expected a lot more detail about your Medline/PubMed search to help put your narrative findings into context. For example, information about the search terms, dates (e.g. if you only looked at articles from a particular year onwards), if you only looked for peer-reviewed articles, only ones in English, how many articles you found, if you screened any out because they weren’t actually relevant or you couldn’t access the full article etc. Without more information, I’ve no idea if you are basing your findings on 2 articles of 20,000, so I have no confidence or ability to decide whether what you found is important or not.
Page 2, line 57 and line 66. A very minor point, but why is it DSM-5 but DSM-III to IV? For consistency it should either be DSM-V or DSM-3 to 4 rather than a mix of digits and roman numerals.
Page 2, lines 71-73. Personal preference, but I think I would find it easier to follow the three delirium levels if they were listed down the page, e.g.
1)
2)
3)
Also, in line 73, rather than talking about ‘features 1 and 2’ and ‘features 3 or 4’ it would make more sense to say ‘features a and b’ and ‘features c or d’ as that’s how you’ve listed them previously. I initially read features 1 and 2 to mean the delirium levels that were actually numbered 1 and 2 so you’re introducing unnecessary confusion for the reader. Please just be a bit more consistent.
Page 2, line 75. I wasn’t going to be highlighting each use of ‘patients’, but I will here. As you’re reporting on articles relating to NHR, surely you should be talking about ‘797 residents’?
Page 2, line 76. I assume you mean ‘researchers’ not ‘researches’
Page 2, line 76. I think the word ‘not’ can be deleted. ‘Not rarely’ would imply ‘often’ and I don’t think you mean that.
Page 2, line 77. Instead of ‘when present’ and ‘when absent’ it would be good to say ‘when the researcher was present’ and ‘when they were absent’. I’d also suggest splitting the sentence after ‘absent’ as it’s currently quite long and difficult to follow. The next sentence could then start ‘This suggests that’
Page 2, line 78. I’d suggest saying ‘enhanced by education nurses in’
Page 2, lines 80-81. I’m not sure I understand the sentence ‘Nursing Home…mimic CAM.’ I don’t actually know what you’re trying to say.
Page 2, lines 80-87. This paragraph raises a question about CAM. If NH-CAM exists, do some nursing homes still use CAM? If yes, that’s fine. If not, why is the previous paragraph about CAM there as you’re supposed to be talking about ‘screening tools used in the NHR’. It’s probably not a problem, but it just made me wonder about CAM and how/where it’s used and why a nursing home would use it rather than the NH-CAM.
Page 2, line 83. The phrase ‘easily distraction’ doesn’t fit. Something like ‘being easily distracted’ or maybe even ‘easy distraction’ at a push might be better.
Page 2, line 86. I’m not sure which one you mean, but I think rather than ‘stratified resident’s risk’ you should use either ‘stratified a resident’s risk’ or ‘stratified residents’ risks’
Page 2, line 90. Is there a word missing in ‘DSM-IV, covers’? If feels like it should be ‘DSM-IV, it covers’ or ‘DSM-IV, NEECHAM covers’.
Page 2, line 97. I don’t think you need ‘the’ after ‘used in’, as ‘used in hospital’ makes sense.
Page 3, line 105. I’m not sure what you mean by ‘skilled nurse facilities’. Is it facilities or settings where skilled nurses work, or something different? Would something like ‘detect delirium when used by skilled nurses’ be better?
Page 3, line 109. Populations, not population.
Page 3, line 116. I’d suggest using either ‘was not more than 15%’ or ‘was at most 15%’
Page 3, lines 122-126. I don’t understand the two sentences ‘However, it should…to 60%’ It feels like they could do with rewording/reordering to help them make sense.
Section 3 is quite long and could benefit from being split into more than one paragraph to make it easier to read and follow.
Table 1 – please be aware that some of the text on the grey lines has a white background, such as the final column for the Perez-Ross study.
Page 7, lines 136-140. I’m assuming these should be footnotes to explain the abbreviations in Table 1, rather than part of the main article text – which is how they appear at present.
Please do a thorough proofread to get rid of the minor points throughout the article. For example on page 7, line 146 it should be ‘predisposing risk factors’, and in line 147 it should be ‘are precipitating risk factors’.
Page 7, line 147. I think it should be ‘suggested that being aged over 65 years’
Page 7, line 159. I’m confused why you’re referring to a study focusing on intensive care units when you’re looking and NHR. I’m guessing that is linked with the following sentence, but as my next comment indicates, I don’t know what that sentence is trying to say.
Page 7, lines 159-162. I’ve read this sentence a few times and can’t work out what it means. Please rewrite it to provide some clarity.
Page 7, line 166. You appear to suggest that lots of things are ‘precipitating risk factors for NHR’ – i.e. risk factors of becoming a nursing home resident. Do you actually mean this, or do you mean risk factors of delirium for NHR? Please clarify/rewrite as necessary.
The whole of section 4 about risk factors is very confusing. It’s one long, dense paragraph that essentially seems to say that anything and everything is identified as a risk factor. I’m not sure what it is actually trying to tell me – that there is/isn’t a consensus around risk factors? It could do with breaking down into shorter paragraphs with each focusing on a different aspect or grouping studies that fit together.
Page 8, lines 189-191. I’m not sure why you are talking about ‘other clinical settings [such] as geriatrics units’ when your focus is NHR. Same point for the sentence in lines 193-195 where you mention ‘well established in different settings’. Please either keep the focus on NHR and remove studies that relate to other settings, or make it clear why those studies are relevant to what you are looking at.
Page 8, line 195. ‘Chair restraints’
Page 8, line 208. You have not expanded ADL and IADL.
In section 5, my comment about the use of ‘patients’ also applies to the use of ‘subjects’. Why are they not just ‘people’ or ‘residents’? It might be what the original studies use, but I think you should change the language to fit the focus of you article.
Section 6 would benefit from being split into more than one paragraph to make it easier to read and follow.
Page 8, lines 221-225. This is a very long, convoluted sentence that would benefit from being split into more than one sentence.
Page 8, lines 225-226. It seems odd to say that ‘it is well established that risk factor[s] which influence the onset of delirium are numerous and heterogeneous’, when you just spent the whole of section 4 making me think that you’ve just done something new by pulling all that information together. I know your focus is (supposed to be) on NHR, but I’m not sure what you are finding/adding that is ‘new’ if you’re saying that it’s already ‘well established’.
Page 8, line 233. Please don’t use the term ‘in elderly’. Something like ‘in nursing home residents’ or ‘in older people’ would be better.
The conclusion feels very superficial and basic, especially considering all the different points you raised. If you are able to summarise your work to one sentence about each area and give a few examples, it would have been useful to have some form of summary at the end of each section. For example when talking about risk factors and telling the reader all of the different ones in the various studies, a short paragraph to pull out the key ones (which I assume is the ones you mention in the conclusion) would have helped to tie it all together and consolidate the information for the reader.
Overall it didn’t really feel like there was a story or ‘narrative’ through the article. It felt more like a series of statements about the different studies that didn’t really fit together or link to each other. For example, on page 8, 233-238 you go from talking about software to a very short statement about supplemental hydration, then onto maintaining hemoglobin. As a reader it is very difficult to constantly switch between topics.
General point – be careful of missing spaces, for example before references (see (19) and (20) in the text), and on page 3, line 106 ‘7cutoff’
Author Response
Reviewer 3
Abstract, line 11. Two minor points about language. Firstly, when you talk about ‘elderly patients’ do you actually mean patients or elderly people more generally? It would be good if you can only use ‘patients’ when you’re talking about someone being treated in a healthcare setting and the context makes this clear. Secondly, instead of saying ‘2-5% of elderly world-wide’ it would be better to say ‘2-5% of elderly people world-wide’. (The second point also applies on page 1, line 45 where this phrase is also used)
REPLY:Thank you for your comment. We mean patients aged over 65 years in hospital and post-acute setting. In abstract we added hospital and post-acute setting and as indicated elderly people world-wide. Please check abstract lines 11-12.
Page 1, line 33. Similar point about the use of ‘patients’. When saying ‘a study including patients’ is it actually patients or people? I don’t know what the study was about , but it might be useful to give some context by saying something like ‘a study including hospital patients’ (or wherever the study was based) to help the reader know what it was about.
REPLY: This was referred to a meta-analysis study where was investigated the role of delirium on survival, hospitalization and dementia among hospitalized or post-acute care patients aged 65 years reference number 8 in the revised version of our manuscript. This information was added in the text. Please check page 2 lines 32-35.
There are other places where the use of ‘patients’ might need to be checked or clarified with some context, but I’ll leave it up to you to go through these rather than pointing out every case.
REPLY:Thank you! We paid attention regarding this point in the text and patients were used when we referred to hospitalization context.
Page 1, line 36. You don’t need both ‘of’ and ‘for’ when you say ‘risk of for mortality’. I would get rid of the ‘for’
REPLY:Apologize. This was a typing error. However in the revised version of our manuscript this was canceled.
Page 2, lines 49-52. The sentence ‘Nevertheless,…measures.’ doesn’t make sense to me. I suggest rewriting it, maybe to something like ‘Nevertheless, delirium often remains undetected and underdiagnosed. Many reasons, such as a lack of education and training for healthcare providers, poor implementation of validated detection tools, and staff attitudes, make it challenging to detect delirium and implement preventive and therapeutic measures.’
REPLY: We rewrote the phrase as indicated. Please check introduction section.
Page 2, lines 52-54. I would have expected a lot more detail about your Medline/PubMed search to help put your narrative findings into context. For example, information about the search terms, dates (e.g. if you only looked at articles from a particular year onwards), if you only looked for peer-reviewed articles, only ones in English, how many articles you found, if you screened any out because they weren’t actually relevant or you couldn’t access the full article etc. Without more information, I’ve no idea if you are basing your findings on 2 articles of 20,000, so I have no confidence or ability to decide whether what you found is important or not.
REPLY: Thank You for the suggestions. We added the Methods paragraph and provided details regarding our search strategy: search terms, reviewers and their role, the time period and language. Please find in the revised version of our manuscript these modifications in page 2 lines 108-121. Please also check results section, regarding the number of the articles. page 2 Review results section.
Page 2, line 57 and line 66. A very minor point, but why is it DSM-5 but DSM-III to IV? For consistency it should either be DSM-V or DSM-3 to 4 rather than a mix of digits and roman numerals.
REPLY:This is a very interesting observation. When the last version of DSM was published, it was reported as DSM-5 not as DSM-V, probably because we expect future DSM -20, etc etc versions. However, given the interesting question we searched how other authors report it, and indeed mainly it is reported DSMIII-IV and DSM-5, within the same articles. For this reason we decided to report it as DSM-5 and the others in roman number as originally reported. Please check also this article: Gibb K, Seeley A, Quinn T, Siddiqi N, Shenkin S, Rockwood K, Davis D. The consistent burden in published estimates of delirium occurrence in medical inpatients over four decades: a systematic review and meta-analysis study.
Page 2, lines 71-73. Personal preference, but I think I would find it easier to follow the three delirium levels if they were listed down the page, e.g.
1)
2)
3)
Also, in line 73, rather than talking about ‘features 1 and 2’ and ‘features 3 or 4’ it would make more sense to say ‘features a and b’ and ‘features c or d’ as that’s how you’ve listed them previously. I initially read features 1 and 2 to mean the delirium levels that were actually numbered 1 and 2 so you’re introducing unnecessary confusion for the reader. Please just be a bit more consistent.
REPLY:The reviewer is right, in the previous version this created confusion. We corrected and reported as indicated by the Reviewer. Please check page 4.
Page 2, line 75. I wasn’t going to be highlighting each use of ‘patients’, but I will here. As you’re reporting on articles relating to NHR, surely you should be talking about ‘797 residents’?
REPLY: Thank you! Now we report patients for hospital setting and residents for nursing homes.
Page 2, line 76. I assume you mean ‘researchers’ not ‘researches’
REPLY: We mean ''research purpose''.
Page 2, line 76. I think the word ‘not’ can be deleted. ‘Not rarely’ would imply ‘often’ and I don’t think you mean that.
REPLY: This is wright. However, this part was canceled.
Page 2, line 77. Instead of ‘when present’ and ‘when absent’ it would be good to say ‘when the researcher was present’ and ‘when they were absent’. I’d also suggest splitting the sentence after ‘absent’ as it’s currently quite long and difficult to follow. The next sentence could then start ‘This suggests that’
REPLY:In the revised version of our manuscript this part was also cancelled.
Page 2, line 78. I’d suggest saying ‘enhanced by education nurses in’
REPLY:As we reported before, in the revised version of our manuscript this part was also cancelled
Page 2, lines 80-81. I’m not sure I understand the sentence ‘Nursing Home…mimic CAM.’ I don’t actually know what you’re trying to say.
REPLY: This part was rewritten more clearly: ''Nursing Home Confusion Assessment Method (NH-CAM) is based on the four CAM features, which are modified using variables and items from Minimum Data Set (MDS) Resident Assessment Protocol (RAP)''. Please check page 4 lines 256-258.
Page 2, lines 80-87. This paragraph raises a question about CAM. If NH-CAM exists, do some nursing homes still use CAM? If yes, that’s fine. If not, why is the previous paragraph about CAM there as you’re supposed to be talking about ‘screening tools used in the NHR’. It’s probably not a problem, but it just made me wonder about CAM and how/where it’s used and why a nursing home would use it rather than the NH-CAM.
REPLY: This is another interesting observation. Following this question, we checked studies which used CAM and resulted that at least twelve studies in the context of nursing homes have applied CAM, from 2003 to 2022, probably because NH-CAM has not been validated against an external reference standard. This information was added in the text.Please check: page4, lines 248-249, and lines 272, page 5 lines 337-338
Page 2, line 83. The phrase ‘easily distraction’ doesn’t fit. Something like ‘being easily distracted’ or maybe even ‘easy distraction’ at a push might be better.
REPLY: This was corrected in: ''being easily distracted''. Line 262.
Page 2, line 86. I’m not sure which one you mean, but I think rather than ‘stratified resident’s risk’ you should use either ‘stratified a resident’s risk’ or ‘stratified residents’ risks’
REPLY:This was modified in ''NH-CAM is a useful tool in estimating delirium related prognosis among NHR''.
Page 2, line 90. Is there a word missing in ‘DSM-IV, covers’? If feels like it should be ‘DSM-IV, it covers’ or ‘DSM-IV, NEECHAM covers’.
REPLY:That's correct. ''NEECHAM covers''.
Page 2, line 97. I don’t think you need ‘the’ after ‘used in’, as ‘used in hospital’ makes sense.
REPLY: Thank you! ''The'' was cancelled , please check page 5 line 355.
Page 3, line 105. I’m not sure what you mean by ‘skilled nurse facilities’. Is it facilities or settings where skilled nurses work, or something different? Would something like ‘detect delirium when used by skilled nurses’ be better?
REPLY: In this point we mean skilled nurses, referring to nursing homes. This part was rephrased as indicated by the Reviewer: ''when used by skilled nurses''. Please check page 5, line 367.
Page 3, line 109. Populations, not population.
REPLY: Thank you! In the revised version of our manuscript we corrected ''population'' with ''populations'' as appropriate.
Page 3, line 116. I’d suggest using either ‘was not more than 15%’ or ‘was at most 15%’
REPLY: This was corrected, ''not more than 15%''. Please check page 2 line 139.
Page 3, lines 122-126. I don’t understand the two sentences ‘However, it should…to 60%’ It feels like they could do with rewording/reordering to help them make sense.
REPLY: In the revised version of our manuscript, this part was cancelled. However, we wanted to say that differences related to prevalence data across different studies may be related to differences in sample size and population characteristics. This was rephrased and placed in discussion section, page 23, lines 658-660.
Section 3 is quite long and could benefit from being split into more than one paragraph to make it easier to read and follow.
REPLY: Following the reviewer advice the paragraph was modified. Please check page 2 from lines 126.
Table 1 – please be aware that some of the text on the grey lines has a white background, such as the final column for the Perez-Ross study.
REPLY:Thank You! The background of the table was modified (all white), as we used track changes you can still see it, but if the changes will be accepted it will become homogenous.
Page 7, lines 136-140. I’m assuming these should be footnotes to explain the abbreviations in Table 1, rather than part of the main article text – which is how they appear at present.
REPLY: Yes the Reviewer is right. You can find it there since the layout of the pages was modified from vertical to horizontal for the table 1. We added explanation that this is the table legend .
Please do a thorough proofread to get rid of the minor points throughout the article. For example on page 7, line 146 it should be ‘predisposing risk factors’, and in line 147 it should be ‘are precipitating risk factors’.
REPLY:This was corrected in the latest version of our manuscript. We mentioned: ''risk factors''.
Page 7, line 147. I think it should be ‘suggested that being aged over 65 years’
Reply: This was corrected as indicated by the reviewer. Please check page 22 line 663.
Page 7, line 159. I’m confused why you’re referring to a study focusing on intensive care units when you’re looking and NHR. I’m guessing that is linked with the following sentence, but as my next comment indicates, I don’t know what that sentence is trying to say.
REPLY: We wanted to underline that risk factors and predictive models for delirium outcome have been studied mostly in settings other than nursing homes. We rephrased the sentence, page 22 lines 667-668. Please check also our reply to the next point.
Page 7, lines 159-162. I’ve read this sentence a few times and can’t work out what it means. Please rewrite it to provide some clarity.
REPLY:We rephrased it. Please check page 22, lines 669-672.
Page 7, line 166. You appear to suggest that lots of things are ‘precipitating risk factors for NHR’ – i.e. risk factors of becoming a nursing home resident. Do you actually mean this, or do you mean risk factors of delirium for NHR? Please clarify/rewrite as necessary.
REPLY: Our aim was risk factors for delirium onset in nursing home residents. We modified all this part. Please check also our reply to the next point.
The whole of section 4 about risk factors is very confusing. It’s one long, dense paragraph that essentially seems to say that anything and everything is identified as a risk factor. I’m not sure what it is actually trying to tell me – that there is/isn’t a consensus around risk factors? It could do with breaking down into shorter paragraphs with each focusing on a different aspect or grouping studies that fit together.
REPLY: We agree with the Reviewer concern, andwe rephrased the ''risk factors'' section and summarized risk factors for delirium in nursing homes following an order which corresponds to: demographic, social, comorbidities, and drugs. Age, dementia, physical restraints and falls result the most common triggers for delirium. This sentence was added in discussion section, page 22, lines 672- 674, and please also check conclusions: page 24 lines: 841-844.
Page 8, lines 189-191. I’m not sure why you are talking about ‘other clinical settings [such] as geriatrics units’ when your focus is NHR. Same point for the sentence in lines 193-195 where you mention ‘well established in different settings’. Please either keep the focus on NHR and remove studies that relate to other settings, or make it clear why those studies are relevant to what you are looking at.
REPLY: We made substantial revision to our manuscript, and based in our aim we referred to studies conducted in nursing homes. In discussion section some comparisons where made with other settings, but with the scope to discuss differences and to underline the main point that delirium in nursing homes is understudied. Please check review' results section and discussion section in the new version of our manuscript: pages : 2-5 and 22-23.
Page 8, line 195. ‘Chair restraints’
REPLY: Thank You! this was corrected.
Page 8, line 208. You have not expanded ADL and IADL.
REPLY:Done , we expanded ADL and IADL. Please check lines 762-763.
In section 5, my comment about the use of ‘patients’ also applies to the use of ‘subjects’. Why are they not just ‘people’ or ‘residents’? It might be what the original studies use, but I think you should change the language to fit the focus of you article.
REPLY: All the section was rephrased, we paid attention in using patients and residents as appropriate. Please check page 3 in the new version of our manuscript.
Section 6 would benefit from being split into more than one paragraph to make it easier to read and follow.
REPLY: Thank You! This part was also rewritten and re-organized.Paragraphs regarding multicomponent interventions, educational programs, and other interventions were discussed.
Please check page 16, lines 459-503.
Page 8, lines 221-225. This is a very long, convoluted sentence that would benefit from being split into more than one sentence.
REPLY:We agree. Part of this phrase was replaced in discussion, but only a simple sentence. Please check Discussion section line 644.
Page 8, lines 225-226. It seems odd to say that ‘it is well established that risk factor[s] which influence the onset of delirium are numerous and heterogeneous’, when you just spent the whole of section 4 making me think that you’ve just done something new by pulling all that information together. I know your focus is (supposed to be) on NHR, but I’m not sure what you are finding/adding that is ‘new’ if you’re saying that it’s already ‘well established’.
REPLY: This part was cancelled.In the revised version of our manuscript we underlined the novelty of our review: a) compared to previous important reviews which included in general long-term settings we focused on nursing homes and b) and provided a review of studies reporting data not only on epidemiology and risk factors but also included outcomes and interventions. Furthermore, we discussed the most frequent delirium screening tools used in nursing homes, reporting their utility, validity and suggested new direction for future studies. Please check the revised version of our manuscript page22 lines 645-655; lines 673-684; page lines 750-751, lines 770-780.
Page 8, line 233. Please don’t use the term ‘in elderly’. Something like ‘in nursing home residents’ or ‘in older people’ would be better.
REPLY: We revised this, with residents.
The conclusion feels very superficial and basic, especially considering all the different points you raised. If you are able to summarise your work to one sentence about each area and give a few examples, it would have been useful to have some form of summary at the end of each section. For example when talking about risk factors and telling the reader all of the different ones in the various studies, a short paragraph to pull out the key ones (which I assume is the ones you mention in the conclusion) would have helped to tie it all together and consolidate the information for the reader.
REPLY: Following the reviewer suggestion we modified the conclusions and summarized the relevant findings related to the scope of our review point by point. Please, check Conclusions, page 24 lines 834-851.
Overall it didn’t really feel like there was a story or ‘narrative’ through the article. It felt more like a series of statements about the different studies that didn’t really fit together or link to each other. For example, on page 8, 233-238 you go from talking about software to a very short statement about supplemental hydration, then onto maintaining hemoglobin. As a reader it is very difficult to constantly switch between topics.
REPLY: Thank you for revising our manuscript and the comments. We re-organized, re-wrote and modified our manuscript. We introduced the results section regarding epidemiology, risk factors, outcome, screening and prevention of relevant data in nursing home settings. Furthermor, the results and findings were discussed in discussion section. Delirium in nursing homes is understudied and we hope that the present paper may be useful for researchers. We ask the reviewer to consider HIS/HER decision regarding our paper.
General point – be careful of missing spaces, for example before references (see (19) and (20) in the text), and on page 3, line 106 ‘7cutoff’
REPLY: Apologize, corrected!
We wish to thank the Reviewer for His/Her constructive criticism which helped us to improve our manuscript.

Round 2
Reviewer 1 Report
Thank you very much for taking into consideration all my comments. I think you have done a very hard work and the final result is much better. The article in its current state is very organized, understandable and gives a very clear vision of the current knowledge about delirium in NHR.
I have some minor comments, mainly about typing errors: in line 141 I think you want to say “continued care”, some authors’ names are wrong (at least Sepulveda in line 152, McCusker and Perez-Ros in table 1, but I advise to review carefully all authors).
The only comment I have about content is in the Discussion section. In the second paragraph, you comment about the difficulties in distinguishing delirium and dementia. I totally agree with you that ICD-10 (you have a typo there, since you wrote CAM) and DSM-5 do not provide specific indications about delirium when dementia is co-present. However, dementia is mainly characterized by memory impairments, whilst attention alteration is a cardinal symptom of delirium, even if mild disturbances could be seen also in dementia. Motor, sleep/wake cycle alterations, thought process and language alterations are also core delirium symptoms, and though they could be also present in some degree in patients with dementia but without delirium, the degree of their alteration may help to differentiate both entities, as you may find in the literature.
Author Response
Reviewer#1
Thank you very much for taking into consideration all my comments. I think you have done a very hard work and the final result is much better. The article in its current state is very organized, understandable and gives a very clear vision of the current knowledge about delirium in NHR.
I have some minor comments, mainly about typing errors: in line 141 I think you want to say “continued care”, some authors’ names are wrong (at least Sepulveda in line 152, McCusker and Perez-Ros in table 1, but I advise to review carefully all authors).
REPLY:Thank you for your comments!The typing errors mentioned by the Reviewer were corrected. All authors' names (from the included studies) were revised and correction were made in the tables and text. In addition, we revised the whole text and corrected some other typing errors in different sections.
The only comment I have about content is in the Discussion section. In the second paragraph, you comment about the difficulties in distinguishing delirium and dementia. I totally agree with you that ICD-10 (you have a typo there, since you wrote CAM) and DSM-5 do not provide specific indications about delirium when dementia is co-present. However, dementia is mainly characterized by memory impairments, whilst attention alteration is a cardinal symptom of delirium, even if mild disturbances could be seen also in dementia. Motor, sleep/wake cycle alterations, thought process and language alterations are also core delirium symptoms, and though they could be also present in some degree in patients with dementia but without delirium, the degree of their alteration may help to differentiate both entities, as you may find in the literature.
REPLY: Following the Reviewer's helpful suggestion the paragraph related to dementia and delirium overlapping was rephrased. We underlined the differences, pointing to the less accentuation of some clinical characteristics (as mentioned by the Reviewer) and the progressive onset. References were added as appropriate. Please check page 22, lines 444-448.
Once again, we wish to Thank the Reviewer for His/Her criticism and constructive revision which helped us to improve the quality of our manuscript.

Reviewer 3 Report
Thank you for taking on board my previous comments. I can see that you have put a lot of effort into your revised article. While your article has improved greatly, there are still a few issues which I feel need to be resolved - see below. Many are similar comments to my first review in that they relate to long paragraphs, clarity, accuracy in language and attention to detail, so it's slightly disappointing that the new content didn't take these on board.
P2 lines 122-123. What do you mean by ‘the prevalence of female population ranged from 46.8% to 90.8%’? Do you mean that ‘the prevalence of delirium in females ranged from 46.8% to 90.8%’ or that ‘46.8% to 90.8% of the populations in the studies were female’? It’s not clear.
In my first set of review comments I indicated that some parts were quite dense with long paragraphs that could benefit from being split into several shorter paragraphs. I think this still applies to some of your new paragraphs, e.g. 3.1 could be split, probably before Morichi and also before ‘The incidence of delirium…’. 3.2 is also very long and dense. In your response to my previous comments you said that you’d reordered the risks into an ‘order which corresponds to: demographic, social, comorbidities, and drugs’, so maybe splitting the big paragraph into these separate topics might make sense. Some of the discussion paragraphs are also quite long. Your section 3.5 is a good example of how you present your information.
At the start of 3.2 it might be useful to have a sentence to set the scene, so something along the lines of saying that while many studies were in agreement about risk factors, there was also some contradiction identified. This would lead in nicely to your two examples about age and gender, rather than launching straight into it.
P5 lines 325-327. The sentence ‘In terms of response burden…’ doesn’t really make sense as it is currently written. I think I know what you’re trying to say, but it could do with a bit of rewording, maybe to something like ‘Regular daily administration may be difficult or burdensome, however…’
Please check your use of ‘result’ and ‘resulted’ as in a few places it does not appear to be the appropriate word. I’ve probably suggested some alternatives in the minor points below, but as it is a recurring issue I thought it was worth highlighting. For example, p22 line 671, ‘drugs were identified as risk factors’ might be better, and on line 672 something like ‘falls are common and significant’ could work.
P22 lines 675-676. I’m not sure I understand what you mean by ‘Despite the definition of dementia is variable’ Please consider rewording to clarify this sentence, and again look at the use of ‘resulted’.
Please also look at P22 lines 679-680 as I’m not sure what the sentence means.
P23 lines 792-793. This sentence about dementia and delirium doesn’t really fit with this paragraph. I think it would probably go better with the earlier bit about dementia, maybe line 678.
Minor points
P2 line 94. ‘Many reasons such as lack…’
P2 lines 96-97. ‘Hence, it is crucial to understand in depth the knowledge…’ or ‘Hence, it is crucial to have in-depth knowledge regarding…’
P2 line 101. Should it be ‘Two researchers’ or ‘Two authors’ not ‘Two reviewers’? (and also in line 105)
P2 line 102. ‘Combinations’
P2 line 129. You haven’t said what 4AT is
P2 line138. I’d suggest ‘delirium at 49.2%, with 57.9% mainly presenting as…’ Also, it’s worth clarifying if you mean 57.9% of NHR affected by COVID-19, or 57.9% of the 49.2% as it’s currently not clear to me.
P2 line 141. ‘continued care’
P3 line 169. ‘malnutrition such as low…’
P3 line 174. Should it be ‘case study series’?
P3 lines 195-195. I think you’ve already set up the abbreviation for DSM-5, so you don’t need to do it again here. Same with CAM in line 221. And NH-CAM in line 242, NEECHAM later on. The other option is to remove most of the tools from 3.1 and only include those that relate to specific studies.
P4 line 212. ‘do not provide’
P4 line 236. ‘CAM was helpful’
P5 line 338. ‘nursing homes are affected’
P16, line 467. ‘Dehydration was significantly’
P16 line 501. ‘prescription of drugs and the occurrence’
P16 line 504 and 505. I’m still not keen o the use of ‘subjects’. I don’t see why you can’t refer to people as ‘residents’
P21, top line. ‘dehydration’
P22, lines 634-637. Unless you have been told to have each table legend as a separate entity that assumes no prior knowledge, I think the Table 2 legend has lots of unnecessary content – you’ve already said what CAM, NH-CAM, NEEHAM, NHR are previously in the article.
P22 line 640. ‘in the healthcare’
P22 line 641. ‘on the hospital setting’
P22 lines 643-644. It should either be ‘between 1.4% and 70%’ or ‘from 1.4% to 70%’
P22 line 645. ‘since then’ not ‘since than’
P22 line 647. ‘Although we focused’
P22 line 649. It should either be ‘between 10% and 60%’ or ‘from 10% to 60%’
P22 line 654 and P24 line 853. ‘superimposed on dementia’
P23 line 789. ‘admitted into post-acute care’
P23 line 798. ‘homes is not common’
P23 lines 798-799. ‘time-consuming’
P23 line 801. ‘have shown good performance’
P23 line 804. ‘evaluated in-depth’
P23 line 814. ‘conditions such as hip fracture’
P23 line 816. ‘behaviors such as doll therapy’
P23 line 818. Do you mean ‘casual’ or ‘causal’?
P23 line 820. ‘interventions such as’
P24 lines 856-857. I think you probably mean ‘well-known risk factors such as age and dementia, physical restraints and falls should be considered…’. i.e. age and dementia are the well-known risk factors. Physical restraints and falls also need to be considered.
Please also check your article for minor bits such as P2 line 139, it should be ‘NHR: 40%’ not ‘NHR :40%’ and the lack of a space before some references such as (42), (26, 44), (13,41) and others. Some of the commas you use also seem to be in odd places, changing how you read sentences. For example, P23 line 795, I don’t think you need the , after ‘Despite’. There are quite a few cases throughout the article which caused me to have to re-read sentences a few times (e.g. line 667 the , after NHR really confused me as it made the phrase ‘and in other contexts as NHR’ a sub-clause and didn’t make sense). I wasn’t going to mention it, but there were just too many cases to ignore. (e.g. line 859 should be ‘Different delirium detection tools have been developed, however…’)
Author Response
Reviewer#3
Thank you for taking on board my previous comments. I can see that you have put a lot of effort into your revised article. While your article has improved greatly, there are still a few issues which I feel need to be resolved - see below. Many are similar comments to my first review in that they relate to long paragraphs, clarity, accuracy in language and attention to detail, so it's slightly disappointing that the new content didn't take these on board.
P2 lines 122-123. What do you mean by ‘the prevalence of female population ranged from 46.8% to 90.8%’? Do you mean that ‘the prevalence of delirium in females ranged from 46.8% to 90.8%’ or that ‘46.8% to 90.8% of the populations in the studies were female’? It’s not clear.
REPLY: Thank you for revising our manuscript, and for your comments. Regarding your question, 46.8-90.8% of the populations included in the studies were female. We rephrased the sentence to make it clear. Please check page 2, lines 75-76 in the new revised version of our manuscript.
In my first set of review comments I indicated that some parts were quite dense with long paragraphs that could benefit from being split into several shorter paragraphs. I think this still applies to some of your new paragraphs, e.g. 3.1 could be split, probably before Morichi and also before ‘The incidence of delirium…’. 3.2 is also very long and dense. In your response to my previous comments you said that you’d reordered the risks into an ‘order which corresponds to: demographic, social, comorbidities, and drugs’, so maybe splitting the big paragraph into these separate topics might make sense. Some of the discussion paragraphs are also quite long. Your section 3.5 is a good example of how you present your information.
REPLY: Following the Reviewer's comments we re-organized section 3.1 in separate paragraphs regarding prevalence and incidence. Section 3.2 also was re-organized into smaller separate sections regarding different risk factors. Please check page 2 line 77, page 3 line 97; page 3 lines 106, 113, 122 and page 4 lines 149, and 152. Some of the discussion paragraphs were shortened.
At the start of 3.2 it might be useful to have a sentence to set the scene, so something along the lines of saying that while many studies were in agreement about risk factors, there was also some contradiction identified. This would lead in nicely to your two examples about age and gender, rather than launching straight into it.
REPLY: Thank You for the suggestions. This sentence was added in our manuscript. Please check page 3 line 107.
P5 lines 325-327. The sentence ‘In terms of response burden…’ doesn’t really make sense as it is currently written. I think I know what you’re trying to say, but it could do with a bit of rewording, maybe to something like ‘Regular daily administration may be difficult or burdensome, however…’
REPLY: Thank You very much! We modified the sentences as suggested by the reviewer. Please check page 5 line 235.
Please check your use of ‘result’ and ‘resulted’ as in a few places it does not appear to be the appropriate word. I’ve probably suggested some alternatives in the minor points below, but as it is a recurring issue I thought it was worth highlighting. For example, p22 line 671, ‘drugs were identified as risk factors’ might be better, and on line 672 something like ‘falls are common and significant’ could work.
REPLY: We performed modifications regarding the use of result and resulted as indicated by the Reviewer.
P22 lines 675-676. I’m not sure I understand what you mean by ‘Despite the definition of dementia is variable’ Please consider rewording to clarify this sentence, and again look at the use of ‘resulted’.
REPLY: The Reviewer is wright; this is not clear. This part was rephrased please check page 22, lines 457-459.
Please also look at P22 lines 679-680 as I’m not sure what the sentence means.
REPLY: We wanted to say that identification of dementia as a risk factor in NHR is in line with other studies performed in other settings. We rephrased the sentences to make them clear. Please check page 22 lines 460-461
P23 lines 792-793. This sentence about dementia and delirium doesn’t really fit with this paragraph. I think it would probably go better with the earlier bit about dementia, maybe line 678.
REPLY: Please re-consider this point. This part was related to measures and interventions which may be investigated and applied for delirium incidence reduction. However, we rephrased the sentence for making it suitable for the paragraph. Please check: page 23, lines 578-579.
Minor points
P2 line 94. ‘Many reasons such as lack…’
REPLY: Corrected, please check page 2 line 47.
P2 lines 96-97. ‘Hence, it is crucial to understand in depth the knowledge…’ or ‘Hence, it is crucial to have in-depth knowledge regarding…’
REPLY: Corrected, please check page 2 line 49.
P2 line 101. Should it be ‘Two researchers’ or ‘Two authors’ not ‘Two reviewers’? (and also in line 105)
REPLY: Ok, this was replaced with authors.
P2 line 102. ‘Combinations’
REPLY: Corrected, please check page 2 line 55.
P2 line 129. You haven’t said what 4AT is
REPLY: The 4 ''A''s test WAS added, please check page 2, line 84.
P2 line138. I’d suggest ‘delirium at 49.2%, with 57.9% mainly presenting as…’ Also, it’s worth clarifying if you mean 57.9% of NHR affected by COVID-19, or 57.9% of the 49.2% as it’s currently not clear to me.
REPLY: Current studies report that the prevalence of delirium among NHR with COVID-19 disease ranges from 49.2% to 57.9% (references 33 and 38). This was corrected to make more clear the sentence. Please check page 3, line 98.
P2 line 141. ‘continued care’
REPLY: Corrected. Please check page 2 line 105.
P3 line 169. ‘malnutrition such as low…’
REPLY:Corrected. Please check page 3 line 127.
P3 line 174. Should it be ‘case study series’?
REPLY:Corrected, please check page 3 line 137.
P3 lines 195-195. I think you’ve already set up the abbreviation for DSM-5, so you don’t need to do it again here. Same with CAM in line 221. And NH-CAM in line 242, NEECHAM later on. The other option is to remove most of the tools from 3.1 and only include those that relate to specific studies.
REPLY: We removed repeated abbreviations.
P4 line 212. ‘do not provide’
REPLY: Thanks, corrected, page 4 line 188.
P4 line 236. ‘CAM was helpful’
REPLY:Corrected, please check page 5 line 219.
P5 line 338. ‘nursing homes are affected’
REPLY: Corrected, please check page 6 line 273.
P16, line 467. ‘Dehydration was significantly’
REPLY:Corrected, please check page 17, line 295.
P16 line 501. ‘prescription of drugs and the occurrence’
REPLY:Corrected, page 17 line 330.
P16 line 504 and 505. I’m still not keen o the use of ‘subjects’. I don’t see why you can’t refer to people as ‘residents’
REPLY:Apologize, subjects was replaced with residents.
P21, top line. ‘dehydration’
REPLY:Corrected, please check page 22 top line.
P22, lines 634-637. Unless you have been told to have each table legend as a separate entity that assumes no prior knowledge, I think the Table 2 legend has lots of unnecessary content – you’ve already said what CAM, NH-CAM, NEEHAM, NHR are previously in the article.
REPLY: The unnecessary content was removed.
P22 line 640. ‘in the healthcare’
REPLY:Corrected, please check page 22 line 360.
P22 line 641. ‘on the hospital setting’
REPLY:Corrected!
P22 lines 643-644. It should either be ‘between 1.4% and 70%’ or ‘from 1.4% to 70%’
REPLY:Thanks, we corrected between ...and....
P22 line 645. ‘since then’ not ‘since than’
REPLY:Corrected!
P22 line 647. ‘Although we focused’
REPLY:Despite was replaced with Although.
P22 line 649. It should either be ‘between 10% and 60%’ or ‘from 10% to 60%’
REPLY:Corrected, between ..and..
P22 line 654 and P24 line 853. ‘superimposed on dementia’
REPLY: Corrected!
P23 line 789. ‘admitted into post-acute care’
REPLY:Corrected.
P23 line 798. ‘homes is not common’
REPLY:Corrected.
P23 lines 798-799. ‘time-consuming’
REPLY: Time-spending was replaced with time-consuming.
P23 line 801. ‘have shown good performance’
REPLY:Corrected.
P23 line 804. ‘evaluated in-depth’
REPLY: Corrected.
P23 line 814. ‘conditions such as hip fracture’
REPLY:Corrected, please check 576.
P23 line 816. ‘behaviors such as doll therapy’
REPLY:Corrected.
P23 line 818. Do you mean ‘casual’ or ‘causal’?
REPLY:Causal, corrected!
P23 line 820. ‘interventions such as’
REPLY:''Such'' was added.
P24 lines 856-857. I think you probably mean ‘well-known risk factors such as age and dementia, physical restraints and falls should be considered…’. i.e. age and dementia are the well-known risk factors. Physical restraints and falls also need to be considered.
REPLY:Thank You! Corrected. Please check Conclusion section, page 23 and 24.
Please also check your article for minor bits such as P2 line 139, it should be ‘NHR: 40%’ not ‘NHR :40%’ and the lack of a space before some references such as (42), (26, 44), (13,41) and others. Some of the commas you use also seem to be in odd places, changing how you read sentences. For example, P23 line 795, I don’t think you need the , after ‘Despite’. There are quite a few cases throughout the article which caused me to have to re-read sentences a few times (e.g. line 667 the , after NHR really confused me as it made the phrase ‘and in other contexts as NHR’ a sub-clause and didn’t make sense). I wasn’t going to mention it, but there were just too many cases to ignore. (e.g. line 859 should be ‘Different delirium detection tools have been developed, however…’)
REPLY:Thank you for the careful revision, we corrected spaces, typos, commas etc and hope that now all are correct.
Once again, we wish to Thank the Reviewer for His/Her criticism which helped us to improve the quality of our paper.
